# Erythropoietin alleviates lung ischemia-reperfusion injury by activating the FGF23/FGFR4/ERK signaling pathway

Xiaosheng Jin[1], Weijing Jin[2], Guoping Li[1], Jisheng Zheng[1] and Xianrong Xu[1]

[1] Pulmonary and Critical Care Medicine, Tongde Hospital of Zhejiang Province, Hangzhou, China
[2] Department of Neonatology, Hangzhou Children's Hospital, Hangzhou, China

## ABSTRACT

**Background**. The purpose of the present study was to investigate the effect of erythropoietin (EPO) on lung ischemia-reperfusion injury (LIRI).

**Methods**. Sprague Dawley rats and BEAS-2B cells were employed to construct an ischemia-reperfusion (I/R)-induced model *in vivo* and *in vitro*, respectively. Afterward, I/R rats and tert-butyl hydroperoxide (TBHP)-induced cells were treated with different concentrations of EPO. Furthermore, 40 patients with LIRI and healthy controls were enrolled in the study.

**Results**. It was observed that lung tissue damage, cell apoptosis and the expression of BAX and caspase-3 were higher in the LIRI model *in vivo* and *in vitro* than in the control group, nevertheless, the Bcl-2, FGF23 and FGFR4 expression level was lower than in the control group. EPO administration significantly reduced lung tissue damage and cell apoptosis while also up-regulating the expression of FGF23 and FGFR4. Rescue experiments indicated that EPO exerted a protective role associated with the FGF23/FGFR4/p-ERK1/2 signal pathway. Notably, the expression of serum EPO, FGF23, FGFR4 and Bcl-2 was decreased in patients with LIRI, while the expression of caspase-3 and BAX was higher.

**Conclusion**. EPO could effectively improve LIRI, which might be related to the activation of the FGF23/FGFR4/p-ERK1/2 signaling pathway.

Corresponding author
Weijing Jin, wzjwj0228@163.com

## INTRODUCTION

Lung ischemia/reperfusion (I/R) injury (LIRI) frequently occurs in patients with surgery, trauma, and pulmonary embolism, which threatens the health of patients, with high morbidity and mortality (*Ojanguren et al., 2023*). Despite rapid advances in medicine, LIRI still poses a serious burden to patients around the world, due to its complex pathophysiological processes which are characterized by a series of harmful cascade events (*Mehew et al., 2023*; *Omorou et al., 2023*). Therefore, further research into potential mechanisms is crucial for the development of potential therapeutic strategies in the future.

Erythropoietin (EPO), a glycoprotein hormone, is secreted by peritubular interstitial cells and the liver, whose function is to stimulate red blood cell production (*Jelkmann, 2003*), In addition to its hematopoietic activity, EPO, together with its derivatives, of notice, has

been manifested to exert pronounced anti-apoptotic and broad tissue-protective effects against damage in the course of the I/R process, or cytotoxic agents in the brain, the heart, the kidneys, and the liver (*Bhangal et al., 2023*; *Kragesteen et al., 2023*; *Zheng et al., 2023*). Furthermore, EPO and its receptor (EPOR) are functionally expressed in diverse non-hematopoietic organs (*Iso, Usui & Suzuki, 2023*; *Ward et al., 2023*). Significantly, it has been validated that it is characterized by strong tissue-protective properties which are associated with anti-oxidative, -apoptotic, -inflammatory, and -excitotoxic effects, as well as angiogenic and neurogenic effects (*Lin et al., 2022*; *Patel et al., 2021*; *Xue et al., 2016*). However, additional studies and clinical trials are required to assess the role of EPO in LIRI.

The fibroblast growth factor (FGF) system is composed of 18 ligands that emit signals through their specific FGF receptors (FGFRs) to communicate with other cells (*Goetz & Mohammadi, 2013*; *Ornitz & Itoh, 2015*). FGF23 is a bone-derived, phosphaturic hormone known to interact with a variety of organ systems, which is a typical member of the FGF family, mediating numerous cellular processes through the binding and activation of related FGFR and the coreceptor klotho (*Chen et al., 2023*; *Thomas, Li & Faul, 2023*). Typical pharmacological and genetic activation of FGF23 signaling results in hypophosphatemic disorders, while its inhibition leads to hyperphosphatemic disorders (*Ho & Bergwitz, 2021*). FGF23 expression is upregulated during the recovery phase after renal I/R injury, and experimental evidence confirms that reducing FGF23 levels can alleviate renal I/R injury (*Chang et al., 2021*; *Junho et al., 2022*). Therefore, maintaining a balance of FGF23 in the plasma is of utmost importance. EPO is a significant regulator of FGF23 production and cleavage (*van Vuren et al., 2019*). It has been confirmed in animal models that exogenous EPO administration can enhance the expression of FGF23 (*Clinkenbeard et al., 2017*). Moreover, FGF23 can increase the expression of FGFR4 (*Grabner et al., 2017*). The cardiopulmonary phenotype of adult FGFR4 knockout mice has been shown to include emphysema, airway inflammation, and right ventricular hypertrophy. However, the involvement of FGF23/FGFR4 in the LIRI process remains unknown. Therefore, we aim to investigate the role and mechanism of EPO in alleviating LIRI and examine the roles played by FGF23/FGFR4 in this process.

Hence, Sprague Dawley rats and BEAS-2B cells were utilized to establish an ischemia-reperfusion (I/R)-induced model *in vivo* and *in vitro*. The study found that EPO was able to mitigate the injury and apoptosis induced during LIRI in both animal and cell models. Additionally, in serum samples from LIRI patients, the levels of EPO, FGF23, and FGFR4 were all significantly lower than those in healthy samples. Mechanistically, EPO exerts its protective effects on tissues and cells by activating the FGF23/FGFR4/p-ERK1/2 pathway. This study provides a theoretical basis for EPO therapy in treating LIRI and reducing its threat to patients.

## MATERIALS AND METHODS

### Clinical sample

A total of 40 blood samples from patients with LIRI at our hospital were collected in the present study. Inclusion criteria were as follows: (1) all patients were diagnosed with LIRI

**Table 1** Age and gender distribution among LIRI patients and healthy blood donors.

| | LIRI patients ($n = 40$) | Healthy donors ($n = 40$) |
|---|---|---|
| Age | 62.98 ± 6.897 | 59.95 ± 6.782[ns] |
| Male/Female | 24/16 | 24/16 |

**Notes.**
ns, no significant difference between two groups.

for the first time; (2) there were no significant abnormalities in liver and renal function in all patients; and (3) the clinical data of each patient was complete. Exclusion criteria were as follows: (1) patients with a history of chemotherapy or radiation; (2) patients who cannot cooperate; and (3) patients with infectious diseases or malignant tumors. Also, samples were taken from 40 healthy volunteers as control. Table 1 describes the distribution of age and sex between two groups. Each participant and their families were informed and agreed to participate in this study, and they signed a consent form. The human study was approved by the ethics committee of Tongde Hospital of Zhejiang Province (2022-072 (K)) and conducted according to the Declaration of Helsinki.

## Animal model and pharmacological treatment

Sixty six male SD rats (8–10 weeks, weight 200–250 g) were purchased from Beijing SiPeiFu biotechnology Co., LTD. The rats received humane care in accordance with the Institutional Animal Care and Use Committee. All rats were kept in cages with controlled temperature (25 °C, approx.) and a 12-hour light/dark cycle, with food and water *ad libitum*. Before the operation, the rats were fasted for 12 h and water was forbidden for 4 h. The rats were randomly divided into two groups, with nine rats in each group: control group and the left LIRI model group. The specific methods are as follows: (1) SD rats of the experimental group were anesthetized by intraperitoneal injection of 100 mg/kg ketamine +10 mg/kg toluene thiazide; (2) after anesthesia, the rats were placed on a table and placed in a supine position, and the limbs of the rats were fixed on the rat operating table; (3) an anterior median incision was made on the neck to expose and separate the right trachea and carotid artery. The trachea was carefully incised for intubation, and mechanical ventilation was performed by the external small animal ventilator. (4) The chest cavity and the left pulmonary hilus were exposed along the left sternum, and the blocking band then passed under the left sternum. After resting for 5 min, the left pulmonary hilus was blocked with a blocking line at the end of the expiratory breath. Reperfusion was performed 1 h later.

Different treatments were performed in these rats ($n = 8$ in each group). EPO dosage (3,000 U/kg) was established with reference to a previous study (*He et al., 2018*). To explore the dose–response relationship, a low-dose group (1,000 U/kg) has been added. The rats were grouped as follows: control group, EPO single administration group (3,000 U/kg), LIRI model group, LIRI model + EPO prevention group (3,000 U/kg), LIRI model + EPO low dose treatment group (1,000 U/kg), LIRI model + EPO high dose treatment group (3,000 U/kg). The EPO pretreatment group was given caudal vein administration 15 min in advance, the treatment group was given caudal vein administration at the same time of reperfusion, and both the prevention group and the treatment group were given

caudal vein administration every day after modeling. Three days post-reperfusion, the rats were euthanized by $CO_2$ inhalation, and blood, alveolar lavage fluid and lung tissue were collected for follow-up studies. The grouping for *in vivo* experiments is illustrated in Fig. S1. The animal experiment was approved by the Committee of Experimental Animals of Zhejiang Academy of Traditional Chinese Medicine (KTSC2021416).

## Cell culture and treatment

BEAS-2B cells were purchased from the ATCC (Manassas, VA, USA) and cultured in RPMI 1640 medium (Invitrogen, Carlsbad, CA, USA) supplemented with 5% heat-inactivated fetal bovine serum (Gibco, Waltham, MA, USA), two mmol/l l-glutamine (Gibco, Waltham, MA, USA), 1% penicillin-streptomycin (Gibco, Waltham, MA, USA) at 37 °C and 5% $CO_2$ in humidified air. Small interfering RNA (siRNA) oligonucleotides targeting FGF23 (siFGF23), siFGFR4, FGF23 overexpression vector (OE-FGF23), OE-FGFR4 and control vector were bought from GenePharma (Shanghai, China). Transfection was performed by using Lipofectamine 2000 Reagent (Invitrogen, Carlsbad, CA, USA) according to the manual. Tert-butyl hydroperoxide (TBHP)-treated BEAS-2B cells were selected as a biological model to establish the I/R cell model.

## Histological analysis

Morphological changes in the lobar tissue of the left lung were examined using hematoxylin and eosin (HE). Lung tissues collected from the rats with different treatments were fixed in 4% paraformaldehyde overnight at 4 °C and embedded in paraffin for 50–60 min at room temperature. Subsequently, the samples were cut into 3–5 μm slices and stained with HE and the tissue morphology was analyzed. The histological changes were examined blindly *via* a light microscope (BA210T; MOTIC, Xiamen, China) by two independent researchers.

## Immunohistochemistry

The expression level of FGF23 in lung tissue was detected by immunohisochemistry (IHC). In short, 3% hydrogen peroxide was employed to block lung tissues at room temperature for 1 h. Antigen retrieval was performed by microwave irradiation in citrate retrieval solution (PH 6.0). Secondarily, each slice was incubated overnight at 4 °C with rabbit monoclonal anti-FGF23 (ab192497, 1: 1000; Abcam, Shanghai, China). The sections were washed and subsequently incubated with a secondary antibody (ab205718, 1: 2000; Abcam, Shanghai, China) at 37 °C for 30 min. Then the sections were stained with DAB until the stain developed, and nuclei counterstaining was performed with hematoxylin. Finally, the slices were observed under a microscope.

## Enzyme-linked immunosorbent assay

Enzyme-linked immunosorbent assay (ELISA) was employed to determine the release of FGF23 in both the serum and BALF. The collected blood samples from the patients were centrifuged at 1,000 rpm at 4 °C for 10 min to obtain the serum supernatant. The supernatant of the tissue homogenate was collected for further analysis. ELISA kits were used to assess the concentration of EPO (PE230, Beyotime, Suzhou, China), FGF23 (PF300, Beyotime, Suzhou, China) and FGFR4 (SEKH-0178, Solarbio, Beijing, China) in

Table 2 **Primer information.**

| Gene | Sequence (5′–3′) |
|---|---|
| Bcl-2-F | GCCTTCTTTGAGTTCGGTG |
| Bcl-2-R | CCAGCCTCCGTTATCCTGGA |
| Bax-F | GCACCAAGGTGCCGGAACTG |
| Bax-R | GGAGAGGAGGCCGTCCCAACC |
| Caspase-3-F | CAGACAGTGGTGTTGATGAT |
| Caspase-3-R | CGGCATACTGTTTCAGCAT |
| GAPDH-F | GGAGCGAGATCCCTCCAAAAT |
| GAPDH-R | GGCTGTTGTCATACTTCTCATGG |

each sample as per the manufacturer's instructions. The optical absorbance at 450 nm was recorded using a microplate reader (Thermo Labsystems, Philadelphia, PA, USA).

## Western blot and co-IP

RIPA lysis buffer (P0013B, Beyotime, Shanghai, China) plus phosphatase inhibitors (35657, Sigma, Burlington, MA, USA) was employed to isolate the total protein of each group, following the manufacturer's instructions. Subsequently, proteins in cell lysates were subjected to sodium dodecyl sulfate-polyacrylamide gel electrophoresis and then transferred onto a polyvinylidene difluoride membrane (Merck-Millipore, Darmstadt, Germany), in addition to incubating with a primary antibody. Afterward, the membrane was incubated with a secondary antibody (Cell Signaling Technology, Danvers, MA, USA) at room temperature for 2 h. Finally, the blots were identified using enhanced chemiluminescence (Pierce, Rockford, IL, USA) in agreement with the manufacturer's instructions. Primary antibodies were as follows: FGF23 (ab307420, 1:3000), FGFR4 (ab178396, 1:3000), p-ERK1/2 (ab184699, 1:1000). GAPDH (ab8245, 1:5000) was used as loading control. For co-IP, lysates of $1 \times 10^7$ BEAS-2B cells were immunoprecipitated with IP buffer containing IP antibody-coupled agarose beads, and protein-protein complexes were later subjected to Western blot.

## qPCR analysis

qPCR was employed to assess the expression of genes BAX, BCL-2 and caspase-3. Total RNA was extracted from tissue specimens and treated cells using the TRIzol reagent (Thermo Fisher Scientific, Waltham, MA, USA). The RNA concentration was determined by Nano Drop spectrophotometer (DS-11FX, DeNovix, Wilmington, DE, USA). Then, total RNA was further reverse transcribed into complementary DNA through the PrimeScript RT reagent kit (TaKaRa, Shiga, Japan). qPCR was performed using TB Green Premix Ex Taq (TaKaRa, Wilmington, DE, USA) and LightCycler 96 real-time PCR instrument (Roche, Basel, Switzerland). The amplification cycling reactions (45 cycles) were as follows: 7 s at 95 °C, 10 s at 57 °C and 15 s at 72 °C. Relative quantification values of the target genes were standardized according to the comparative threshold cycle ($2^{-\Delta\Delta CT}$). GAPDH was applied as a HOUSEKEEPING GENE. Primer sequences are displayed in Table 2.

### TUNEL assay

TUNEL assay (C1086, Beyotime, Shanghai, China) was implemented to rate the effect of EPO on TBHP-induced BEAS-2B cell apoptosis under the manufacturer's protocol. In brief, 4% paraformaldehyde was applied to fix the treated BEAS-2B cells for 30 min at room temperature and then cultured with 0.3% Triton X-100 for 5 min. Secondarily, 50 μl TUNEL reaction reagent was added to each sample at 37 °C for 1 h. After staining with 0.5 μg/ml of DAPI for 10 min at room temperature. Finally, TUNEL-positive cells were observed with the help of a fluorescence microscope (Beyotime, Shanghai, China).

### Flow cytometry apoptosis analysis

Annexin V-FITC/PI apoptosis detection kit (556547, BD Biosciences, San Jose, CA, USA) was implemented to determine cell apoptosis, as instructed by the manufacturer. In short, resuspended cells were stained with 10 μL Annexin V-FITC and PI in the dark for 15 min. Subsequently, the apoptotic cells were analyzed by flow cytometry (CytoFLEX, Beckman Coulter, Brea, CA, USA).

### Statistical analysis

All the above assays were carried out independently in triplicates. The data were analyzed with the help of GraphPad Prism 7.04 software and displayed as mean ± SD. Statistical differences were performed *via* One-way ANOVA with Tukey's *post hoc* $t$-test for multiple comparisons. $P < 0.05$ was considered significant. $^{*}P < 0.05$, $^{**}P < 0.01$; $^{***}P < 0.001$.

## RESULTS

### Effect of I/R injury on FGF23 levels and apoptosis

Fibroblast growth factors (FGFs) are critical biological regulators which are associated with pathophysiological processes, such as LIRI. Likewise, it has been evinced that exogenous FGF ligands could protect against I/R, which offers potential therapeutic options for patients with LIRI (*Deng et al., 2020*). In order to investigate the role of FGF23 in LIRI, the LIRI rat model was established. As shown in Fig. 1A, alveolar and interstitial inflammation and pulmonary fibrosis were observed in the lung tissues of rats in the I/R group (model), confirming the successful construction of the LIRI rat model. Next, IHC and ELISA were performed to analyze the level of FGF23 in the lung tissue, the serum, and the BALF sample *in vivo*. It was observed that the FGF23 levels were decreased in lung tissue in the I/R group, in comparison with the control group (Fig. 1B). Also, the data of the ELISA validated similar results in the serum and BALF samples (Figs. 1C & 1D). In addition, Western blot demonstrated that the expression of FGF23, FGFR and p-ERK1/2 was down-regulated in lung tissue in the experimental group compared to the control group (Fig. 1E). Of note, based on the evidence that apoptosis is one of the important phenotypes in the process of ischemia-reperfusion injury (*Bi, Li & Zhang, 2023*), apoptosis-related gene expression and cell apoptosis was detected by qPCR and TUNEL assay. It was indicated that the expression level of Bcl-2 was decreased in the experimental group than in the control group, however, the expression level of BAX and caspase-3 was increased in the group (Fig. 1F). Equally importantly, TUNEL-positive cells were higher in rats with IR injury compared to those of the control group (Fig. 1G).

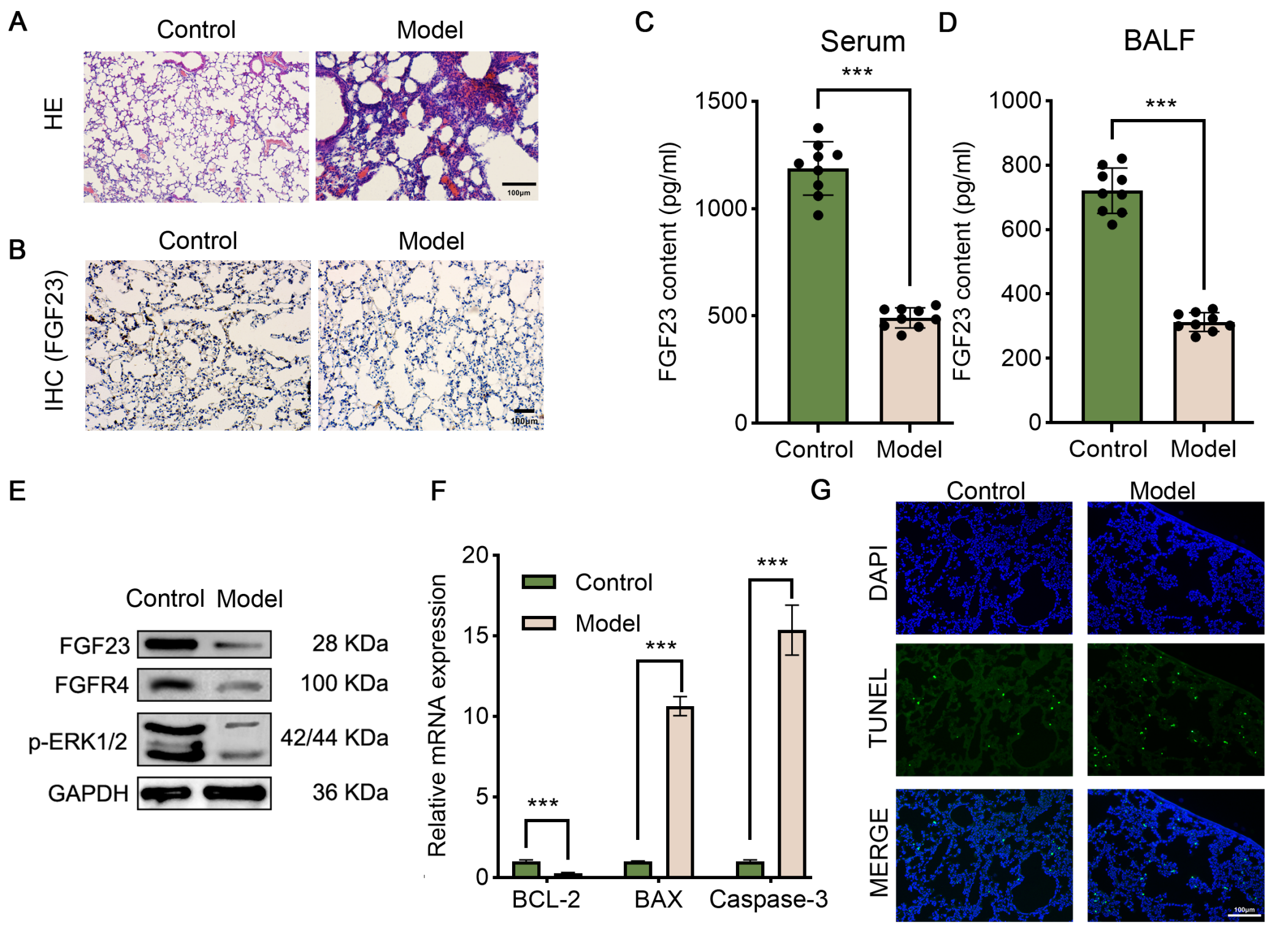

**Figure 1** **The pathologic changes, molecular changes and apoptosis of the LIRI model.** (A) Lung tissues were stained by HE assay to show the morphologic changes. Scale bar, 100 μm. (B) IHC assay was employed to analyze FGF23 content in the lung tissue of rats ($n = 3$). Scale bar, 100 μm. (C & D) ELISA was executed to compare FGF23 content in serum (C) and BALF samples (D) ($n = 9$). (E) Western blot was utilized to measure the expression level of FGF23, FGFR4, p-ERK1/2 in lung tissue ($n = 3$). (F) qPCR was adapted to detect the expression of BCL-2, BAX and caspase-3 mRNA ($n = 3$). (G) TUNEL assay was performed to evaluate TUNEL-positive cells ($n = 3$). LIRI, lung ischemia-reperfusion injury; HE, hematoxylin–eosin; IHC, immunohistochemical staining. ***$P < 0.001$.

## EPO attenuates I/R-induced lung injury *in vivo*

It has been demonstrated that EPO exerts a protective effect on the multiple I/R model (*Kartal et al., 2023*; *Kittur et al., 2023*). However, additional studies and clinical trials are required to assess the role of EPO in LIRI. The results of HE revealed that I/R treatment significantly induced lung injury in comparison with the sham group, which could be partially weakened by pre- or post-treatment of EPO (Fig. 2A). Interestingly, the results of IHC and ELISA showed that FGF23 expression level in lung tissue (Fig. 2B) and content in serum (Fig. 2C), in addition to FGF23 BALF (Fig. 2D) was significantly increased in the EPO administration group compared with the model group. Further, this increased trend appears in a dose-dependent manner (Figs. 2B–2D). In addition, in comparison with the model group, the number of TUNEL-positive cells was decreased in the EPO treatment group (Fig. 2E). Likewise, the qPCR analysis showed that pro-apoptosis related genes BAX and caspase-3 were significantly downregulated along with a decrease in the levels of anti-apoptosis related protein BCL-2 after EPO administartion. (Fig. 2F). In addition, I/R treatment significantly suppressed the expression of FGF23, FGFR4 and

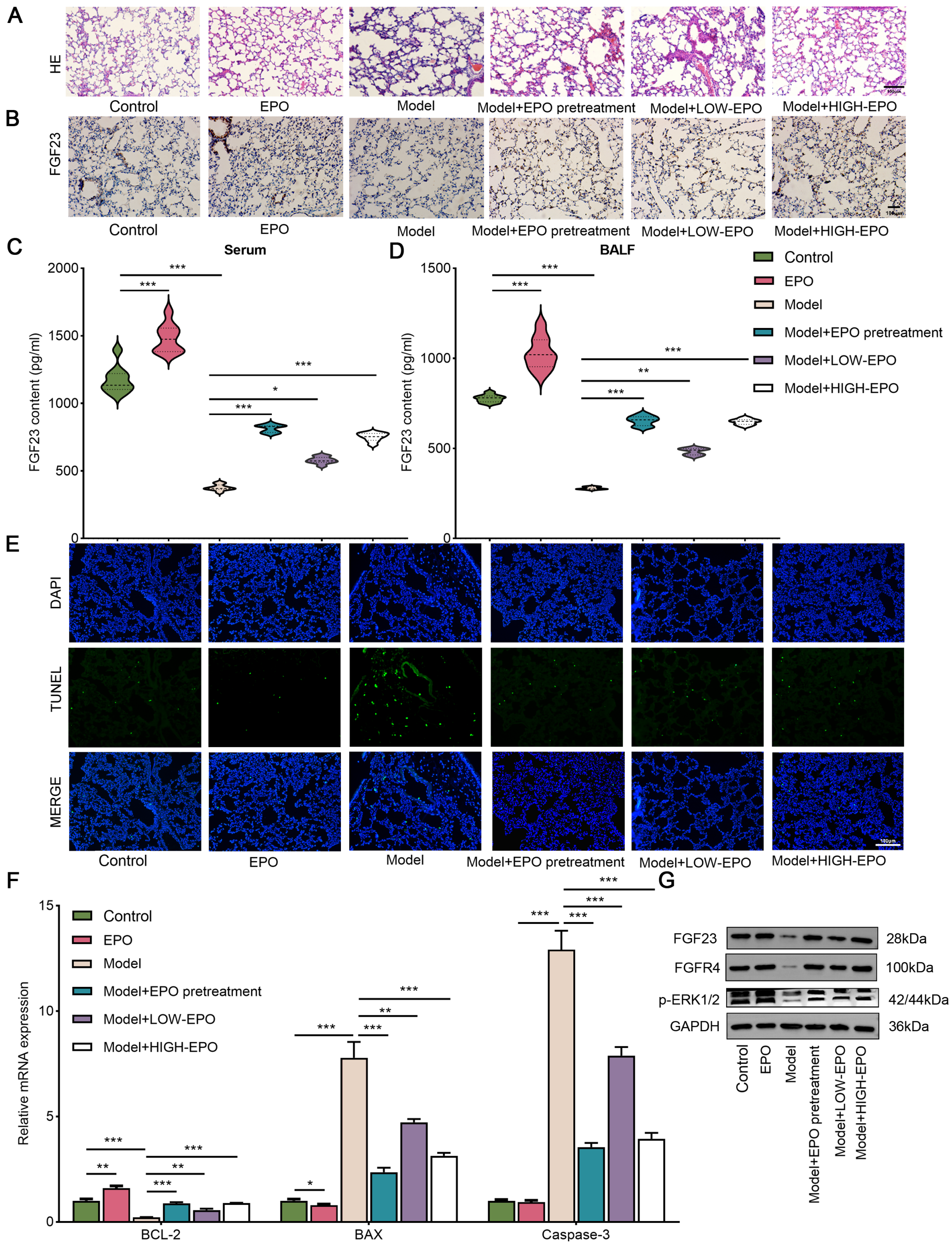

**Figure 2 EPO displays remarkable anti-apoptotic and broad tissue-protective effects against damage triggered by I/R injury.** (A) HE staining was used to detect lung injury in the different groups ($n = 3$). (B) The FGF23 expression level in different groups was valued by IHC ($n = 3$). (C) FGF23 content in serum (C) and BALF sample (D) was estimated by ELSH assay ($n = 9$). (E) Cell apoptosis was evaluated *via* TUNEL assay ($n = 3$). (F) The apoptosis-related gene (including BCL-2, BAX, and caspase-3) were analyzed by qPCR ($n = 3$). (G) Western blot was utilized to measure the expression level of FGF23, FGFR4, p-ERK1/2 in lung tissue ($n = 3$). EPO, erythropoietin; HE, hematoxylin–eosin; IHC, immunohistochemical staining. $^*P < 0.05$; $^{**}P < 0.01$; $^{***}P < 0.001$.

p-ERK1/2, which could be abrogated by EPO pretreatment or posttreatment (Fig. 2G). Additionally, in comparison with the control group, the EPO group did not cause damage to lung tissue and inhibited apoptosis, indicating EPO had no side effect on the rat lung (Figs. 2A–2G). Taken collectively, EPO attenuated I/R-induced lung injury *via* regulating the FGF23/FGFR4/p-ERK1/2 pathway.

### EPO attenuates the injury of TBHP-induced BEAS-2B cells by activating FGF23/FGFR4/p-ERK1/2 pathway

Since EPO has a pronounced promotive activity against injury induced by TBHP in rat lungs, BEAS-2B cells were cultured with different concentrations of TBHP to mimic an I/R injury. Cell apoptosis in the 100 μM TBHP group, 200 μM TBHP group and 400 μM TBHP group was significantly increased compared with the control group (Figs. 3A & 3B). In addition, the results of ELISA demonstrated that TBHP diminished the FGF23 content in cell supernatant in a dose-dependent manner compared with the control group (Fig. 3C). Also, the mRNA expressions of caspase-3, BAX, and Bcl-2 were detected by qPCR, as described (Fig. 3D). The results indicate that TBHP treatment effectively reduced the expression of BCL-2, and increased the expression of caspase-3 and BAX. The expression levels of FGF23, FGFR4 and p-ERK1/2 were also examined by Western blotting (Fig. 3E). It was observed that TBHP remarkably decreased the expression of FGF23, FGFR4 and p-ERK1/2 in BEAS-2B cells (Fig. 3E). In addition, the relationship between FGF23 and FGFR4 was verified by CO-IP assay, which revealed that both FGF23 and FGFR4 proteins were identified by western blot in the immune complexes obtained with anti-FGF23 or anti-FGFR4 antibodies, indicating an interaction between FGF23 and FGFR4 could be disturbed by TBHP treatment. Because the cell model constructed at 400 μM TBHP worked best, it was used in a series of subsequent experiments. In order to verify the protective effect of EPO on TBHP-induced apoptosis of lung epithelial cells, TBHP (400 $\mu$M) was added to cells with EPO (0, 5, 10 U/mL) 30 min later, and cells were collected at different time points for FITC-Annexin V/PI. As displayed in Figs. 4A and 4B, EPO blocked the development of cell apoptosis in comparison to the control group. Similar to the results of the flow cytometry, EPO significantly reduced the expression of BAX and caspase-3 as well as increased the expression of BCL-2 mRNA in BEAS-2B cells treated with TBHP (Fig. 4C). Moreover, EPO could not only enhance the expression of FGF23 and FGFR4 but also strengthen the interaction between them (Figs. 4E & 4F). In addition, EPO could induce the expression of FGF23 in BEAS-2B cells treated with TBHP (Fig. 4E).

### FGF23 or FGFR4 inhibited TBHP-induced apoptosis

Afterward, BEAS-2B cells with FGF23 were either knocked down or overexpressed to investigate the function of FGF23 in cell apoptosis and the transfection outcome was assessed by Western blot (Figs. S2A & Fig. S2B). The FGF23 expression in siFGF23 (BEAS-2B cells transfected with siFGF23) group was strikingly down-regulated compared to the siNC group (BEAS-2B cells transfected with control vector). Flow cytometry analysis showed that transfection with siFGF23 significantly triggered apoptosis compared with

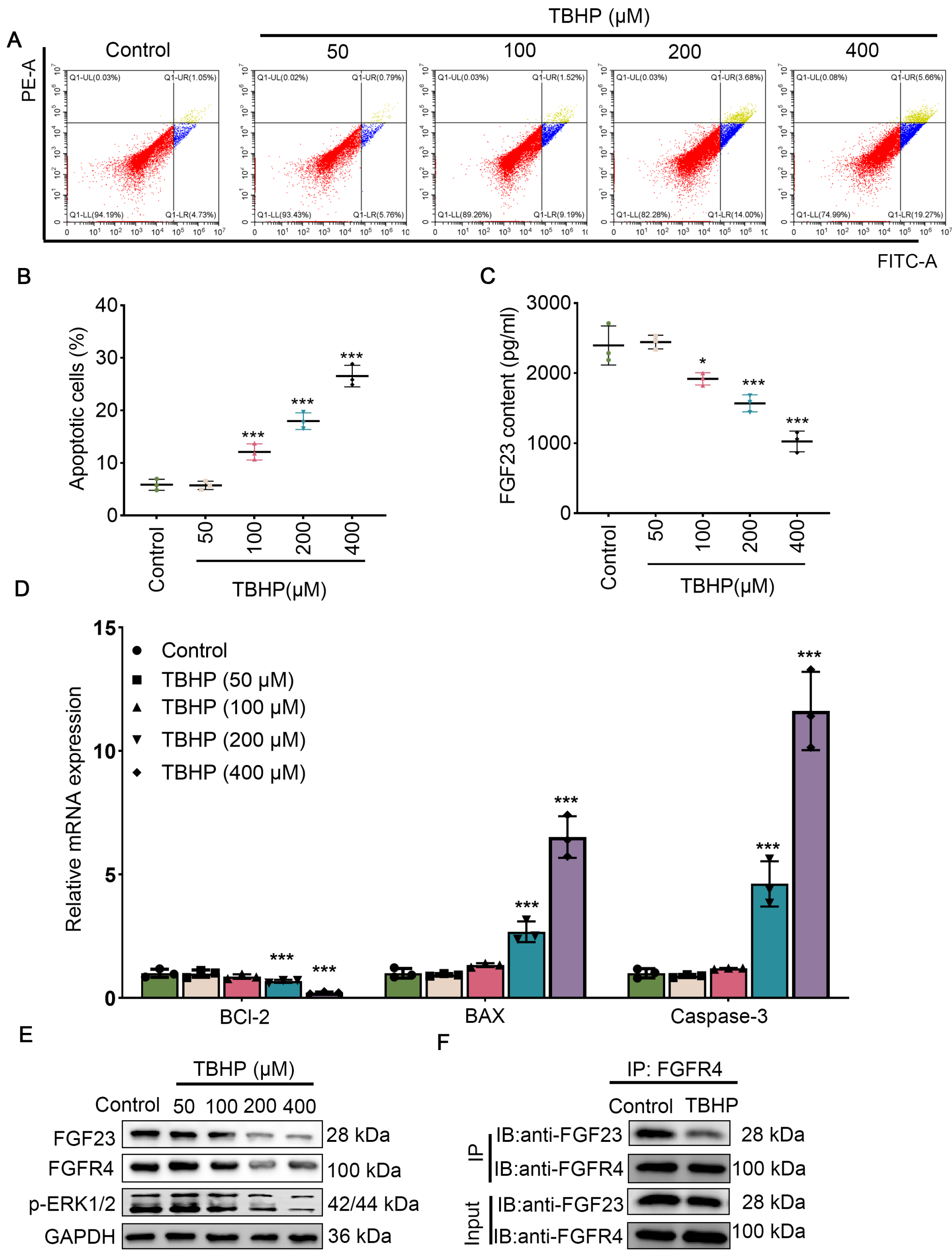

**Figure 3 Different concentrations of TBHP induced BEAS-2B apoptosis.** (A&B) Flow cytometry was employed to evaluate cell apoptosis after treatment with different dose concentrations of TBHP (50, 100, 200, and 400 μm). (C) FGF23 content in the cell supernatant was rated by ELISA upon treatment with different dose concentrations of TBHP (50, 100, 200, and 400 μm). (D) qPCR was performed to confirm the expression level of the apoptosis-related gene (including BCL-2, BAX, and caspase-3) after treatment with different dose concentrations of TBHP (50, 100, 200, and 400 μm). (E) Western blot was exploited to monitor the expression level of FGF23, FGFR4, and p-ERK1/2 in lung tissue. (F) The relationship between FGF23 and FGFR4 was estimated by CO-IP assay after treatment with TBHP or not. EPO, erythropoietin; TBHP, tert-butyl hydroperoxide TBHP. *$P < 0.05$, **$P < 0.01$; ***$P < 0.001$.

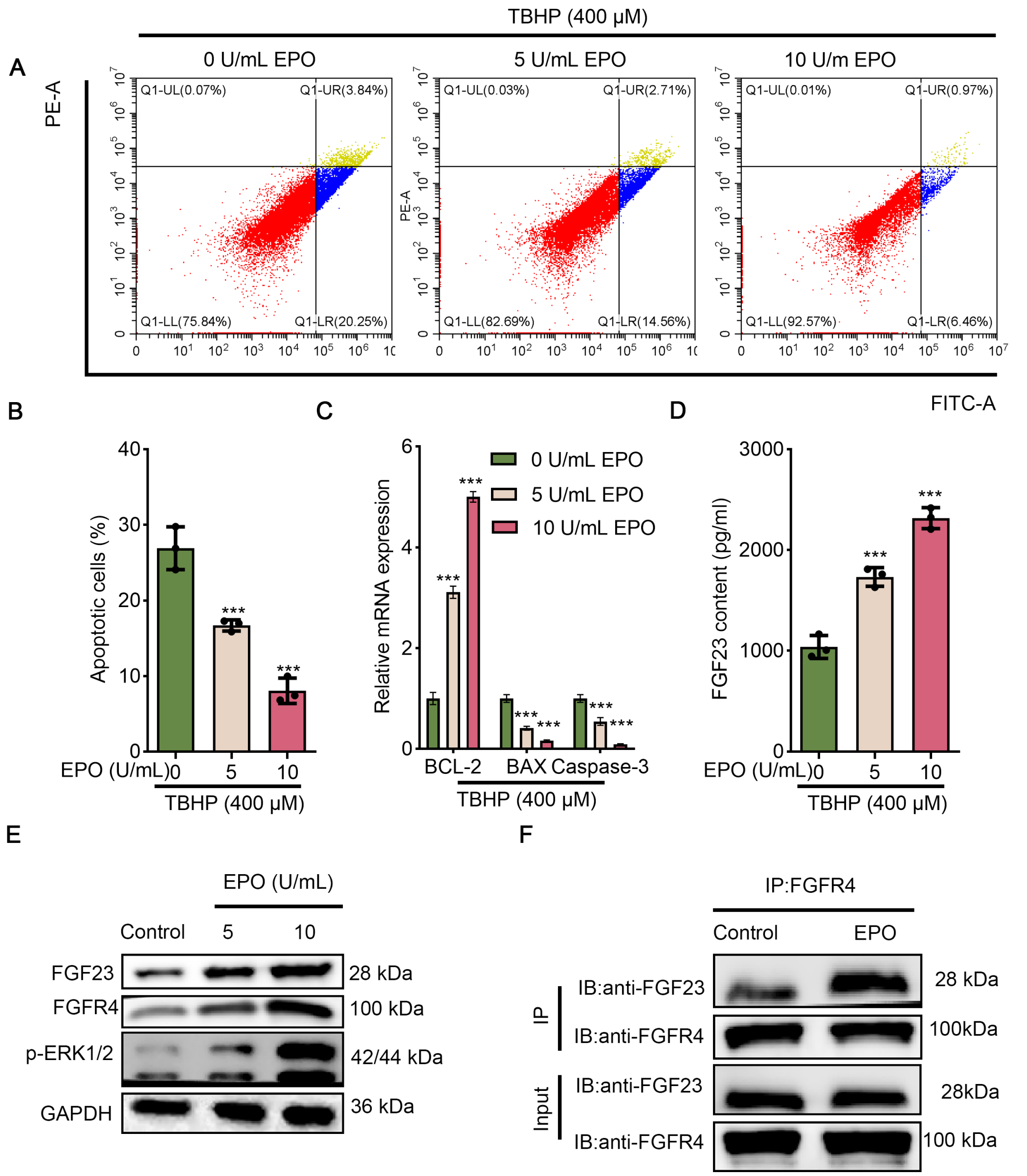

**Figure 4 EPO exerts anti-apoptotic effects in TBHP-induced BEAS-2B cells.** (A & B) Cell apoptosis was measured by flow cytometry in 400 μm TBP-induced BEAS-2B after treatment with different doses of EPO (0, 5, and 10 μm). (C) The apoptosis-related gene (including BCL-2, BAX, and caspase-3) was assessed by qPCR in 400 μm TBP-induced BEAS-2B after treatment with different doses of EPO (0, 5, and 10 μm). (D) FGF23 content in the cell supernatant was estimated by ELISA in 400 μm TBP-induced BEAS-2B after treatment with different doses of EPO (0, 5, and 10 μm). (E) Western blot was exploited to monitor the expression level of FGF23, FGFR4, p-ERK1/2 in 400 μm TBP-induced BEAS-2B after treatment with different doses of EPO (0, 5, and 10 μm). (F) The interaction between FGF23 and FGFR4 was estimated by CO-IP assay after treatment with EPO or not. TBHP, tert-butyl hydroperoxide TBHP. ***$P < 0.001$.

transfection with siNC (Figs. 5A& 5B). Furthermore, the results of ELISA demonstrated that knockdown of FGF23 restricted the FGF23 content in cell supernatant (Fig. 5C). Also, the data of qPCR depicted that FGF23 silencing effectively reduced the expression of Bcl-2, and increased the expression of caspase-3 and BAX (Fig. 5D). As can be easily understood, compared with OE-NC-TBHP group, overexpression of FGF23 significantly inhibited apoptosis and up-regulated the content of FGF23 in cell supernatant (Figs. 5A–5D). Interestingly, knocking down FGFR4 or overexpressing FGFR4 has a similar effect on apoptosis as knocking down FGF23 or overexpressing FGF23 (Figs. S2A & S2B, Figs. 5A–5D). It is worth noting that FGF23 positively regulates the expression of FGFR4 in BEAS-2B cells or cell supernatant; however, regulating the expression of FGFR4 has no significant effect on the expression of FGF23 (Fig. S2).

### EPO exerts an anti-damage effect depending on the regulation of the FGF23 signaling pathway

To validate the interaction between EPO and FGF23 signaling pathway in TBHP-induced lung cells, siFGF23, siFGFR4, FGF23 overexpression plasmid (OE-FGF23), OE-FGFR4 and their control vector were transfected into BEAS-2B cells, respectively. Transfection efficiency was then examined by Western blot (Fig. S3A). Knocking down the expression of FGF23 or FGFR4 promoted apoptosis, enhanced the expression of BAX and caspase-3, and inhibited the expression of Bcl-2 in TBHP-induced BEAS cells in the background of EPO compared with NC group (Figs. 6A–6C). Conversely, overexpression of FGF23 or FGFR4 produced protective effects similar to EPO in TBHP-induced models (Figs. 6A–6C). Furthermore, Western blot results demonstrated that FGF23 or FGFR4 positively regulated the expression of p-ERK1/2 (Fig. S3A). However, up-regulating or down-regulating FGFR4 expression did not significantly affect FGF23 expression, either in BEAS-2B cells or in supernatant (Fig. S3B).

### The key factors of expression level in patients with LIRI

Furthermore, samples from 40 patients with LIRI and 40 healthy controls were collected in the present study to substantiate the role of the above key factors in LIRI. Initially, we compared the difference in EPO levels between the control group and the LIRI group. As shown in Fig. 6A, the results of ELISA revealed that the EPO content in serum from the patients with LIRI was lower than the patients from the control group. Also, the FGF23 and FGFR4 content were decreased compared with the control group (Figs. 7B & 7C). As can be easily understood, the results of Western blot showed that the expression levels of EPO, FGF23 and FGFR4 in LIRI serum were significantly lower than those in healthy controls (Fig. 7D). In addition, the expression levels of caspase-3 and BAX were elevated in comparison with the control; however, the BCL-2 expression level was lower than in the healthy sample (Fig. 7E).

## DISCUSSION

The LIRI process involves I/R rapidly activating inflammation and injury responses, leading to life-threatening edema, organ dysfunction, and cell death (*Tang et al., 2022*;

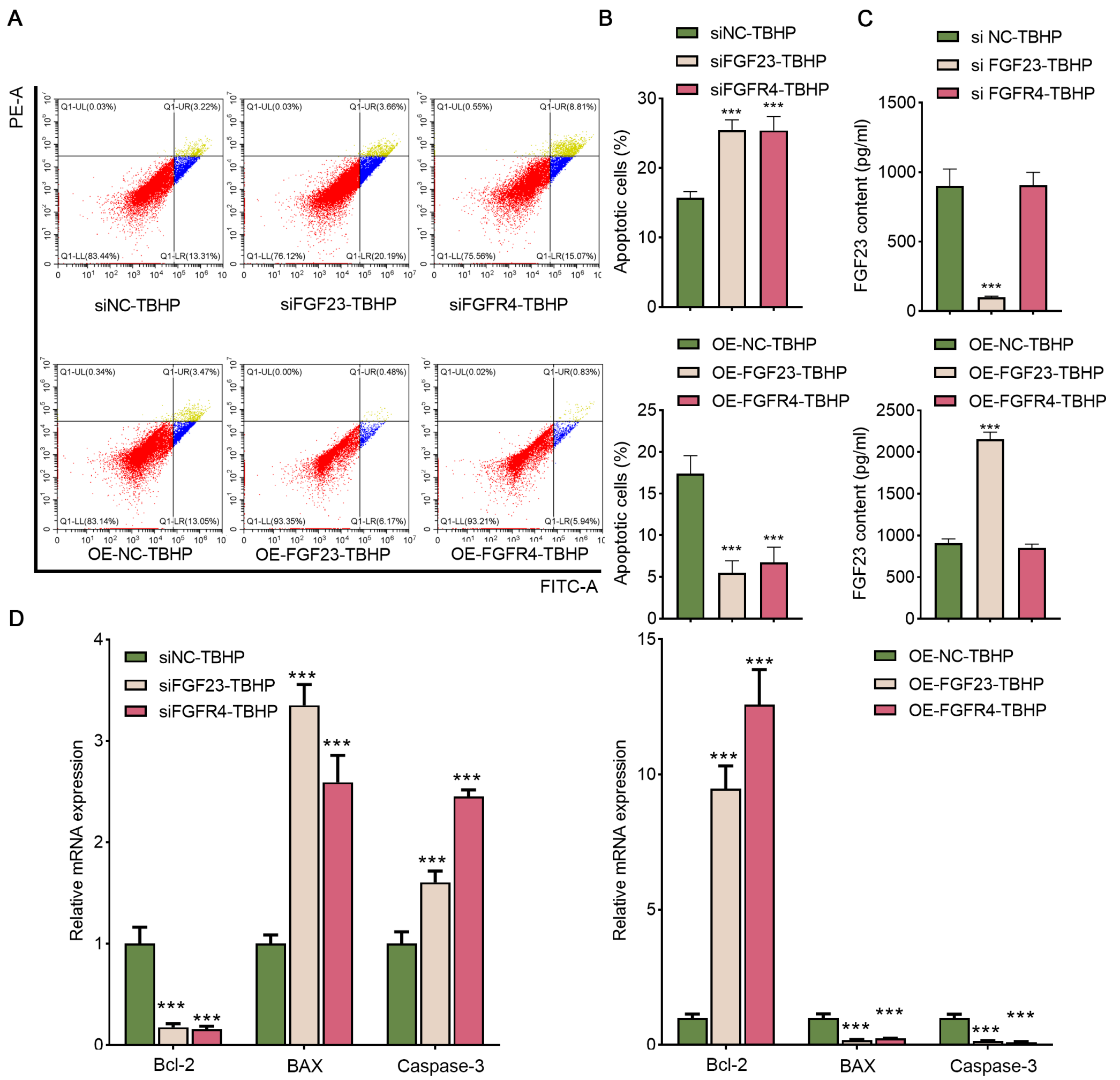

**Figure 5 FGF23 and FGFR4 depressed the apoptosis of TBHP-induced BEAS-2B cells.** (A & B) After siFGF23, siFGFR4, OE-FGF23 and OE-FGFR4 were transfected into the TBHP-induced BEAS-2B cells, cell apoptosis was appraised by flow cytometry. (C) FGF23 content in the cell supernatant was estimated by ELISA after up-regulating and down-regulating FGF23 and FGFR4. (D) The apoptosis-related gene (including BCL-2, BAX, and caspase-3) was assessed by ELISA after up-regulating and down-regulating FGF23 and FGFR4. TBHP, tert-butyl hydroperoxide TBHP. ***$P < 0.001$.

*Vlastos et al., 2022*). Because ischemic preconditioning involves mechanical injury, drug preconditioning is a more ideal treatment method. It is not only non-invasive and safe, but most important is that it can effectively reduce LIRI, so it has a broad clinical application prospect (*Xiao et al., 2022*). The findings of the present study demonstrate that EPO could exert a protective effect against I/R-induced injury, as evidenced by inhibiting cell apoptosis and the expression of pro-apoptosis-related proteins, as well as promoting the expression of anti-apoptosis-related proteins. The beneficial effects of EPO on lung I/R injury involved activation of FGF23/FGFR4/p-ERK1/2 pathway.

EPO, a drug to promote cell generation, has been widely used in clinical applications (*Muras-Szwedziak, Pawłowicz-Szlarska & Nowicki, 2023*; *Skrifvars et al., 2023*). The

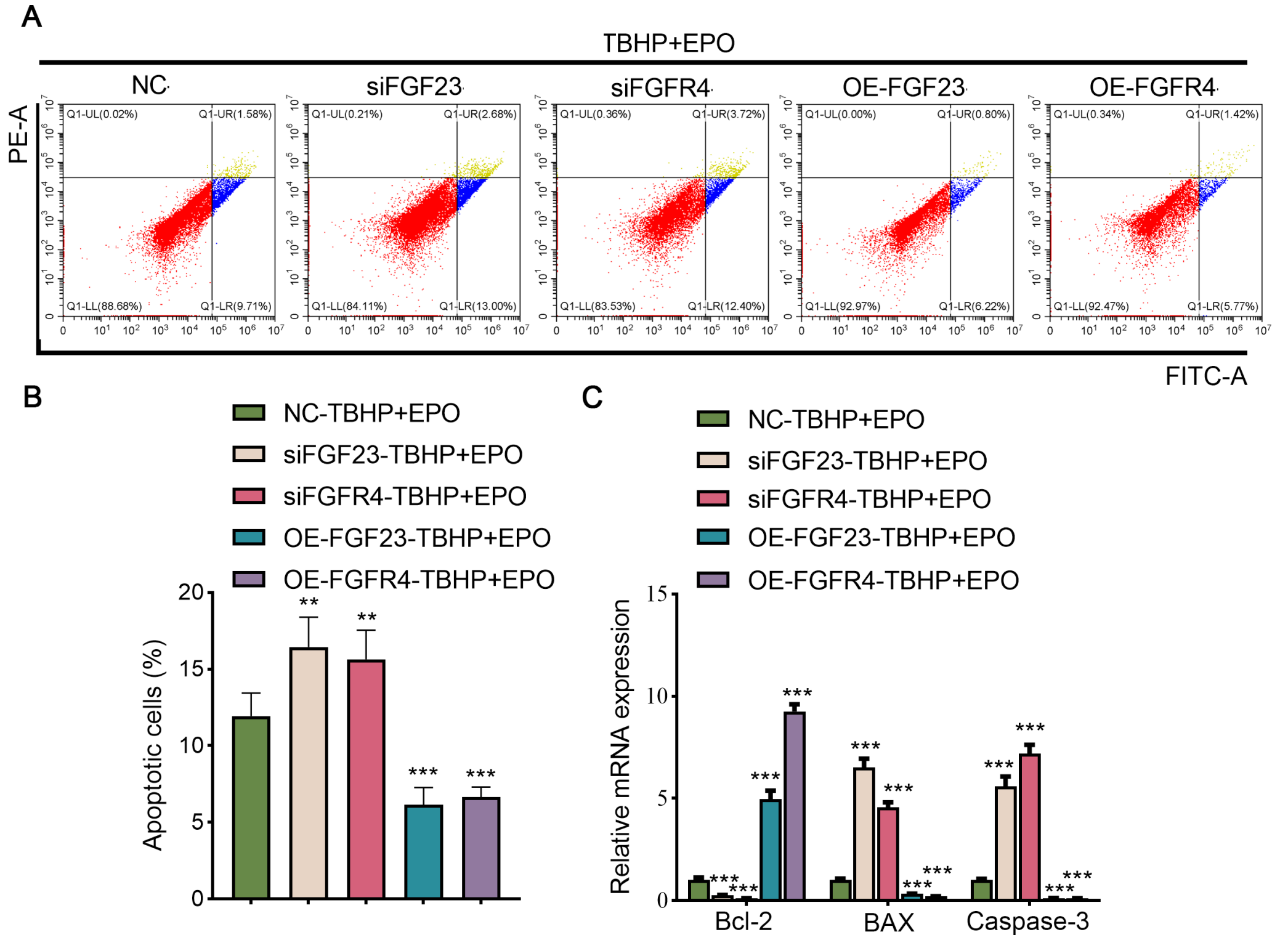

**Figure 6 The anti-apoptotic effect of EPO depends on the regulation of FGF23 pathway.** (A & B) After up-regulating or down-regulating the expression of FGF23 or FGFR4 in BEAS-2B cells co-treated with TBHP and EPO, apoptosis was detected by flow cytometry. (C) qPCR was adapted to estimate the apoptosis-associated genes after cell apoptosis. EPO, erythropoietin; TBHP, tert-butyl hydroperoxide TBHP. $^{**}P < 0.01$; $^{***}P < 0.001$.

function of EPO is to combine with EPOR to promote the differentiation and apoptosis of red blood cells (*Ling et al., 2023*). In addition, EPO is a transmembrane protein composed of multiple amino acids, which is not only expressed on the erythrocyte membrane but also on the surface of other tissue cells, such as neurons, vascular smooth muscle cells, astrocytes, endothelial cells and cardiomyocytes (*Wang et al., 2023*). EPO can be combined with EPOR and help to activate a variety of cytokines to inhibit inflammation, stabilize cell ultrastructure, and anti-apoptosis (*Govindappa et al., 2023*). For example, in cardiomyocytes, EPO has a good protective effect on cardiomyocytes, can effectively reduce the inflammatory response caused by MIRI, reduce infarct size, and promote neovascugenesis, which has a promising prospect in clinical application (*Zafiriou et al., 2014*). A clinical study involving 54 patients demonstrated that EPO can improve lung function and reduce systemic inflammation to alleviate lung injury induced by cardiopulmonary bypass after cardiac surgery (*Lin et al., 2020*). This increases our confidence in studying EPO for treating LIRI. Patient blood samples have provided us with the biological relevance of our study. In the present study, it was observed that the EPO content in serum from the patients with LIRI was lower than health control, suggesting

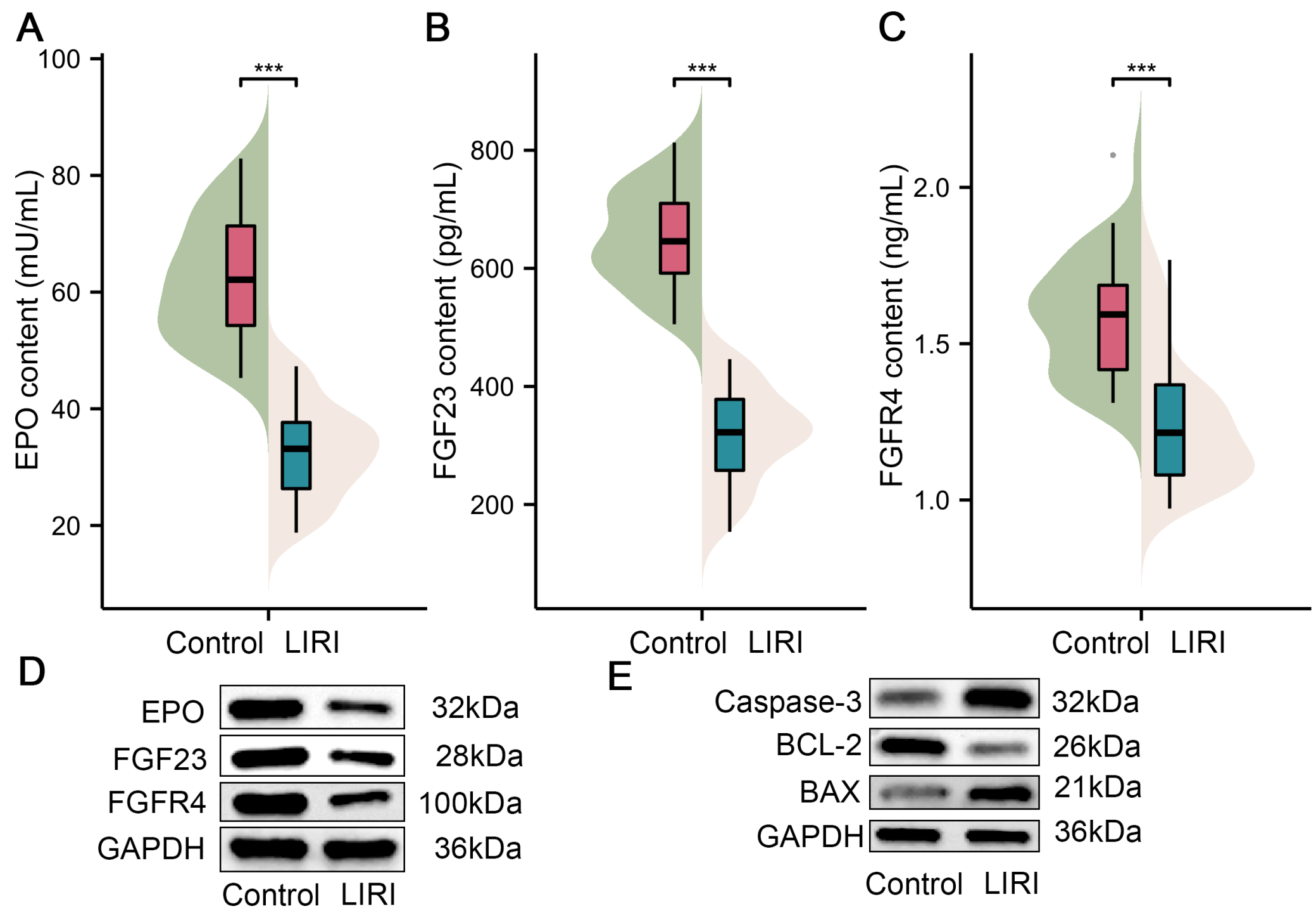

**Figure 7 The expression of key genes in the patients with LIRI.** (A–C) EPO content (A), FGF23 content (B), and FGFR4 content (C) were in the serum of patients with LIRI and health control appraised by ELISA. (D) The expression of EPO, FGF23 and FGFR4 were monitored in the serum of patients with LIRI and health control appraised by Western blot. (E) The expression of caspase-3, BCL-2 and BAX were evaluated in serum of patients with LIRI and health control appraised by Western blot. EPO, erythropoietin; TBHP. ***$P < 0.001$.

that the deficiency of EPO may have adverse effects. By analyzing these blood samples, we can gain a deeper understanding of the characteristics of EPO in LIRI.

The most important manifestation of cell damage is apoptosis, which is the orderly and autonomous death of cells under the control of genes (*Lu et al., 2012*). In order to comprehensively investigate the role of EPO *in vivo*, the LIRI rats model was established to provide simulations of the overall physiological environment. From the results, perspective, both preoperative and postoperative EPO treatments could alleviate the injury induced by I/R. Likewise, TUNEL test showed that EPO can inhibit cell apoptosis induced by LIRI. WB and qPCR tests showed that the expression of pro-apoptosis-related proteins or genes (BAX and caspase-3) was significantly inhibited and the expression of anti-apoptosis-related proteins or genes (BCL-2) was increased after EPO intervention. Interestingly, EPO had no side effects on the rat lung tissue. Besides, TBHP is an oxidant for I/R injury in various cultured cells (*Li et al., 2023*; *Tan et al., 2020*; *Zhuo et al., 2021*). Based on the relatively well-defined mechanisms of TBHP action, TBHP-treated BEAS-2B cells were selected as a biological model to establish the I/R cell model. Cell experiment results indicate that EPO can alleviate apoptosis in I/R cell model, with a concentration-dependent gradient. Those data indicated that EPO might be a therapeutic clue for LIRI treatment.

Prior research has recovered that pretreatment with EPO can alleviate acute lung injury induced by I/R either by inhibiting the Toll-Like Receptor-4/Nuclear Factor-$\kappa$B pathway or by blocking p38 MAPK signaling (*He et al., 2018*; *Jia et al., 2021*). However, what distinguishes this study from previous research is that EPO treatment occurred both before and after the onset of I/R. In other words, this study explores the preventive and therapeutic effects of EPO on LIRI. More importantly, it unveils a distinct underlying mechanism. In the present study, it was observed that FGF23 content was decreased in the patients with LIRI compared to the control group. Cell experiments confirm that EPO can increase the expression of FGF23 and FGFR4 in BEAS-2B cells, and overexpressing FGF23 and FGFR4 can inhibit TBHP-induced apoptosis. Furthermore, in order to explore the interaction between FGF23 and FGFR4 during the process of EPO-mediated apoptosis relief, rescue experiments indicate that regulating the expression of FGFR4 does not affect the expression of FGF23, while increasing the expression of FGF23 enhances FGFR4 expression. This suggests that FGF23 acts as an upstream factor for FGFR4, meaning that EPO induces FGFR4 through FGF23. In summary, mechanistic studies suggest that EPO exerts a protective effect on I/R cell model by positively regulating the FGF23/FGFR4/ERK pathway. Previous studies revealed that elevated levels of FGF23 induce left ventricular hypertrophy in mice through FGFR4 (*Han et al., 2020*). The activation of FGFR4 by FGF23 typically requires the co-receptor α-klotho, while klotho-independent signaling in the patient's body only occurs under conditions of extremely high FGF23 concentrations (*Nakano, Kishimoto & Tokumoto, 2023*). Therefore, patients experiencing cardiac-related diseases during LIRI should closely monitor the circulating FGF23 to avoid potential harm to the body.

In conclusion, the present study indicated that EPO has protective effects against lung injury induced by renal ischemia-reperfusion, but it did not cause side effects. Moreover, the interaction mechanism of EPO with FGF23/FGFR4/ERK signaling contributes to understanding the processes of LIRI and provides new insights for the treatment of lung damage induced by I/R. The toxicology, pharmacodynamics, and detailed mechanism of the protective effect of EPO on patients with LIRI remain to be elucidated by clinical trials in the future.

## Funding

This study was funded by the Basic Public Welfare Research Program of Zhejiang Province (LGF22H010015). The funders had no role in study design, data collection and analysis, decision to publish, or preparation of the manuscript.

## Grant Disclosures

The following grant information was disclosed by the authors:
Basic Public Welfare Research Program of Zhejiang Province: LGF22H010015.

## Competing Interests

The authors declare there are no competing interests.

## Author Contributions

- Xiaosheng Jin conceived and designed the experiments, performed the experiments, authored or reviewed drafts of the article, and approved the final draft.
- Weijing Jin performed the experiments, analyzed the data, authored or reviewed drafts of the article, and approved the final draft.
- Guoping Li performed the experiments, analyzed the data, prepared figures and/or tables, and approved the final draft.
- Jisheng Zheng performed the experiments, analyzed the data, prepared figures and/or tables, and approved the final draft.
- Xianrong Xu performed the experiments, analyzed the data, prepared figures and/or tables, and approved the final draft.

## Human Ethics

The following information was supplied relating to ethical approvals (i.e., approving body and any reference numbers):

The Biomedical Research Involving Humans was approved by ethics review committee of the Tongde Hospital of Zhejiang Province (2022-072 (K)) on March 18, 2022. Written informed consent for this research was obtained from the patients prior to research.

## Animal Ethics

The following information was supplied relating to ethical approvals (i.e., approving body and any reference numbers):

Protocols for animal experiments were approved by the Committee of Experimental Animals of Zhejiang Academy of Traditional Chinese Medicine (KTSC2021416) on July 23, 2021, in compliance with the National Institutes of Health guidelines for the care and use of laboratory animals.

## Data Availability

The raw measurements are available in the Supplementary Files.

## Supplemental Information

Supplemental information for this article can be found online at http://dx.doi.org/10.7717/peerj.17123#supplemental-information.

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
