# Peer review of "Erythropoietin alleviates lung ischemia-reperfusion injury by activating the FGF23/FGFR4/ERK signaling pathway"

_PeerJ, doi:10.7717/peerj.17123_

## Round 0.1 · original submission · Major Revisions

The authors must carefully address all the reviewers' comments.

**Language Note:** The review process has identified that the English language must be improved. PeerJ can provide language editing services - please contact us at copyediting@peerj.com for pricing (be sure to provide your manuscript number and title). Alternatively, you should make your own arrangements to improve the language quality and provide details in your response letter. – PeerJ Staff

Reviewer 1 ·

Basic reporting

1) Authors advised to get a language editing service.
2) Authors should make some changes in the terminology they used (Some are indicated in the annotated manuscript).
3) The authors should improve the introduction section. We should be informed about FGF23, FGFR4, EPO, etc in this section. Also, we should see the hypothesis of authors and the aim of the study.
4) Authors should avoid discussing the outcomes of their experiments or giving explanatory infromation in the results section. In this part, authors should only write their results; other parts could be added to the introduction or discussion sections.
5) Authors should improve the discussion section. The parts highlighted in yellow in the annotated manuscript are only parts that actually belong to a discussion section. In discussion, we should see what your results mean, and we should see the similarities and dissimilarities with the previous studies in this field. Authors are advised to rewrite this section.

For further inquiries, please see the attached pdf file.

Experimental design

1) Regarding the doses used in the in vivo experiments: Please mention how you decide on these doses by citing the research if you adepted it from an earlier study. If you did not use a previous research to determine the high and low doses, please briefly indicate how you decided on these doses for high and low dose concentrations.

2) Please mention the antigen retrieval method and if you performed a counter-staining with hematoxylin to stain the nuclei in the IHC.

3) Regarding the qPCR:
-Which total RNA quantification method was used (nanodrop? Qubit? Etc?)?
-Please mention the analyzed genes in the text, and state which gene was used as the housekeeping gene.
-Please mention the ROX kit that you used.
-Please mention the used real-time PCR device.

For further inquiries, please see the attached pdf file.

Validity of the findings

1) What is the sample size for the experiments you performed? When I check the supplementary data provided for the images, it looks like authors used 6 samples for Fig1, 18 for Fig2, 5 for Fig3, 3 for Fig4, 9 for Fig 5 (both for OE and S,) with three technical repeats for all samples in their ELISA experiments. And for qPCR experiments, it looks like authors used three technical repeats of one sample for each gene in each different experimental group. Is it correct?

2) There are some images with irregularities in the supplementary data, especially for the Fig1E, FGFR4 images for WB. Could you please provide the details on the used protocol for gel preparation and electrophoresis, if you used a phosphatase inhibitor or not, how did you choose the membrane for WB and blotting?

3) Please check the reverse primer sequence of the GAPDH PCR primer, if you wrote it correctly.

4) Please state the reason and importance of using blood samples from patients, and using these in vivo and in vitro models. Authors should make the reader understand why they used these materials, why they made comparisons between these groups, and the significance of the outcomes in the discussion section.

For further inquiries, please see the attached pdf file.

Additional comments

Authors had investigeted the effect of erythropoietin with a focus on FGF23/FGFR4/ERK signaling pathway. Although authors utilized many techniques, and used results they obtain from both in vivo and in vitro experiments, I believe they should improve their writing. Besides some technical questions about the used methodology, the manuscript should be improved especially for the discussion section.

Annotated reviews are not available for download in order to protect the identity of reviewers who chose to remain anonymous.

Reviewer 2 ·

Basic reporting

Overall comments:

The amount of work and evidence presented in this manuscript is extremely impressive

the authors should have a professional editor / medical writer to conduct thorough review of the manuscript. There are multiple grammatical mistakes, spelling typos, and many instances with inappropriate wording choices

The citation format in the manuscript also seems a bit off, especially with the placement of period before the referenced article. Can the authors double check on this too?

There seems to be plenty publication on EPO’s role during LIRI. I’m including a few examples below from a 1min google search. The authors need to do a better job reviewing published articles in this topic and highlighting what is unique / different from in this manuscript compared to the published ones. An alternative approach would be to clearly state that this article aims to pressure test / provide additional evidence on the protective effect of EPO in LIRI. Personally I am not against publishing articles to validate prior work, but this needs to be clearly stated.

https://www.ncbi.nlm.nih.gov/pmc/articles/PMC7191960/

https://www.ncbi.nlm.nih.gov/pmc/articles/PMC5842661/

https://journals.sagepub.com/doi/10.1177/09603271211043480?icid=int.sj-full-text.similar-articles.1

Rats and mice are different – there are multiple instances where the authors used mice instead of rats

How the animals were grouped for the experiment is confusing and conflicting within the method section. To simplify this, the authors may consider using a table or figure to explain the experiment setup?

There is no clear rationale of why the authors selected FGF23, FGFR, and ERK. This need to be laid out in the Introduction section or first paragraph of results section

Experimental design

N/A

Validity of the findings

The conclusion in row 189 is not supported by the evidence – which is merely showing FGF23 decreased in LIRI and apoptosis occurred during LIRI. There are many other pathways that would be involved so I wouldn’t link these two observations together yet

Conflicting data in 2C & 2D vs 1C & 1D --> I thought FGF23 level was reduced in model group? Figure 2C & 2D also do not support the finding descriptions in the 2nd paragraph of Results section. This is concerning and I would prefer to have the authors do a thorough data QC before reviewing the rest of the manuscript.

Additional comments

N/A

---

## Round 0.2 · Major Revisions

All the reviewer's comments must be carefully addressed as recommended.

Reviewer 1 ·

Basic reporting

Thank you for your improvements. There are just a few typos in the manuscript, which could be easily corrected.

Experimental design

Could authors provide a table for the age and gender distribution of the blood donors?

Validity of the findings

The original blot (given in the supplementary files) for Fig1E, FGFR4, looks unusable.

Additional comments

The original blot (given in the supplementary files) for Fig1E, FGFR4, looks unusable.

---

## Round 0.3 · Major Revisions

Please address the remaining concerns of the reviewer and re-run the experiment to assess FGFR4 protein expression.

Reviewer 1 ·

Basic reporting

N/A

Experimental design

N/A

Validity of the findings

It would be great if the authors could re-run the western blot analyses for the questionable blots

Additional comments

It would be great if the authors could re-run the western blot analyses for the questionable blots

---

## Round 0.4 · accepted · Accept

All remaining concerns of the reviewer were addressed and the revised manuscript is acceptable now.

Reviewer 1 ·

Basic reporting

Authors had replied positively to my concerns and re-run the blots.

Experimental design

No comments.

Validity of the findings

Authors had replied positively to my concerns and re-run the blots.

Additional comments

Authors had replied positively to my concerns and re-run the blots.